# Digging into contrastive learning for robust depth estimation with diffusion models

Submission Id: 2846

## ABSTRACT

Recently, diffusion-based depth estimation methods have drawn widespread attention due to their elegant denoising patterns and promising performance. However, they are typically unreliable under adverse conditions prevalent in real-world scenarios, such as rainy, snowy, etc. In this paper, we propose a novel robust depth estimation method called D4RD, featuring a custom contrastive learning mode tailored for diffusion models to mitigate performance degradation in complex environments. Concretely, we integrate the strength of knowledge distillation into contrastive learning, building the 'trinity' contrastive scheme. This scheme utilizes the sampled noise of the forward diffusion process as a natural reference, guiding the predicted noise in diverse scenes toward a more stable and precise optimum. Moreover, we extend noise-level trinity to encompass more generic feature and image levels, establishing a multi-level contrast to distribute the burden of robust perception across the overall network. Before addressing complex scenarios, we enhance the stability of the baseline diffusion model with three straightforward yet effective improvements, which facilitate convergence and remove depth outliers. Extensive experiments demonstrate that D4RD surpasses existing state-of-the-art solutions on synthetic corruption datasets and real-world weather conditions. The code for D4RD will be made available for further exploration and adoption.

## CCS CONCEPTS

• **Computing methodologies** → **Reconstruction**.

## KEYWORDS

Depth Estimation, Robust Perception, Self-supervised Learning, Diffusion Methods

## 1 INTRODUCTION

Estimating accurate depth from a single image is highly attractive because this task arises whenever the 3D structure is needed. Since the supervised Monocular Depth Estimation (MDE) methods[24, 25] require high-cost ground truth (GT) depth labels. To address this limitation, researchers have explored self-supervised approaches that leverage adjacent frame pose information and photometric consistency from video sequences. However, most existing self-supervised MDE models can only handle clear and ideal settings but are highly unreliable under challenging conditions. This limitation significantly hampers the practical applicability of these methods.

Recently, diffusion models have demonstrated their superiority in MDE[14, 23] with remarkable performance. Moreover, some works from other domains[18, 22] have revealed that the denoising diffusion process allows the potential of robust perception. These findings have inspired us to explore how diffusion models can address the challenges faced in MDE and enhance robustness. To

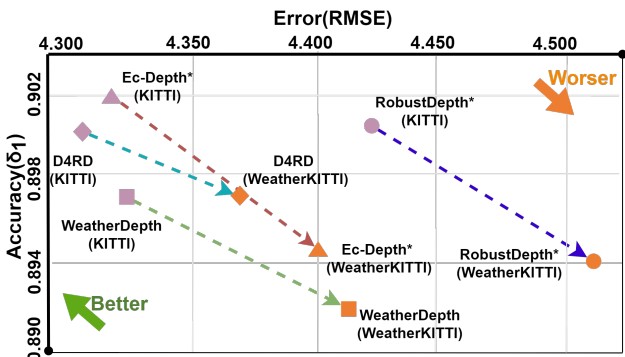

**Figure 1: Comparisons with RMDE method[26, 30, 31] on KITTI and WeatherKITTI. The length of the Line represents the performance degradation magnitude under adverse environments. All the methods are trained on the same dataset WeatherKITTI for fairness.**

this end, we propose **D**iffusion for(**4**) **R**obust **D**epth **(D4RD)**, a novel diffusion-based MDE framework. D4RD incorporates a new multi-level 'trinity' contrastive scheme designed to enhance robustness and mitigate performance degradation in challenging conditions.

To better understand the concept of the 'trinity', we need to revisit the previous efforts aimed at resolving the challenges of Robust Monocular Depth Estimation (RMDE). Generally, these approaches fall into two main categories: contrastive learning-based alignment approaches[19, 30, 31, 35] and knowledge distillation-based pseudo-supervised methods[9, 21, 26]. Considering that corruption does not affect the physical scene depth, the former approach typically involves the clear image along with its various augmentations into the network, enforcing consistency among different output depths to provide extra supervision. However, a common issue of these methods is that directly maximizing the depth similarity can easily fall into a collapsing solution[4]. To address this issue, additional guidance is required, which poses a challenge in a self-supervised manner since the photometric consistency cannot provide a perfect convergence anchor.

The latter usually employs a pre-trained teacher model to estimate the depths of clear images and uses these estimates as pseudo-labels to train the student model on augmented images. Obviously, there is a performance upper bound for the student model as the teacher model is not entirely accurate.

In short, we urgently need a perfect label that can not only serve as an anchor for guiding optimization in contrastive learning but also eliminate performance bottlenecks in knowledge distillation. This label seems impossible in self-supervised learning but we ingeniously find it in the diffusion process—the sampled noise. In the forward diffusion process, we obtain noisy images by adding

a sampled noise that follows a normal distribution to the original image. During the training process, sampled noise is considered as the GT label for supervising noise prediction. When the prediction is accurate, the original image will be accurately restored[11, 29]. Therefore, we incorporate this noise as supplementary guidance for contrastive learning to establish a 'trinity' contrast pattern. Concretely, we maintain the original consistency constraints unchanged, but on this basis, we apply sampled noise constraints to clear/augment input noise predictions, gathering the pair of estimated noise towards a more stable and precise optimum. If the model is completely robust and accurate, then the three types of noise(*i.e.*, clear/augmented/sampled noise) should be identical, that is, the 'trinity'.

Moreover, [13] have found that the burden of robust dense perception is mainly concentrated on a specific component of the network (*e.g.*, the lower layers), which limits the network potential. Based on this, we intuitively try to distribute these responsibilities across the network by extending the 'trinity' paradigm to more generic feature and image levels. Naturally, finding suitable reference points like sampled noise is challenging on these occasions, so we rely on clear scenes and teacher models to provide guidance to the model. Despite that, we found our trinity is still better than the traditional distillation and contrast methods. In addition, to further explore how multi-level contrastive learning promotes network performance, we visualize the perceptual results after each level. The improvement, as depicted in Fig. 4, is likely attributed to each level addressing distinct aspects of image enhancement, ultimately leading to more robust and accurate depth results.

D4RD takes MonoDiffusion [28] as the baseline framework. Before incorporating the multi-level trinity contrastive paradigm, we also enhance the stability of the baseline with three simple but effective improvements. It strengthens convergence and eliminates depth outliers, which lay the foundation for handling complex scenarios. Finally, extensive experiments show that our proposed method, D4RD, achieves SoTA performance among all RMDE competitors across **two** syntheses dataset and **five** real weather conditions. Moreover, as shown in Fig. 1, D4RD not only surpasses other methods with a significant margin but also demonstrates the minimum performance degradation in challenging conditions. In summary, the contributions of this paper are as follows:

- We enhance the diffusion-based depth estimation architecture with better stability and convergence, and further establish a novel robust depth estimation framework, named **D4RD**, with a customized contrastive learning scheme for diffusion models.
- Benefiting from the natural guidance of sampled noise, we ingeniously integrate the strength of distillation learning into contrastive learning to form a 'trinity' contrast pattern, which facilitates noise prediction accuracy and robustness in D4RD.
- The noise-level trinity is then expanded to more generic feature and image levels, building the multi-level trinity scheme. It evenly distributes the pressure of handling domain variances across different model components.

## 2 RELATED WORK

### 2.1 Monocular Depth Estimation

MDE aims to predict the accurate depth from a single image, which is an ill-posed problem. Intuitively, this challenge was initially seen as a dense regression task[6, 7] based on GT depth labels. Four years later, Fu et al.[8] discovered that reconstructing MDE into a depth bins classification task can significantly improve performance. Still about four years later, as the diffusion model showcases its talents in generative task[11, 29], detection[3], and segmentation[32], DDP[12] first reformulates this task as a depth map denoising task, and lead to giant progress again. Followers like Diffusiondepth[5], DDVM[27], VPD[37], TAPD[15], EcoDepth[23], Marigold[14], and MonoDiffusion[28] all demonstrate the advantages of this paradigm in various MDE sub-tasks. D4RD also adopts this promising scheme and extends it on robust depth estimation.

### 2.2 Robust Depth Estimation

As mentioned before, although MDE is an ill-posed problem, with the development of research and task optimization in the past decade, MDE on clear data sets has achieved excellent performance. Therefore, for RMDE tasks, an intuitive and effective method is to use the results of clear scenes, which can be divided into two major categories, knowledge distillation and contrastive learning.

Distillation-based methods focus on leveraging the well-predicted depth map (typically estimated from clear scenes) as pseudo-labels to assist in training the RMDE model. MD4all[9] trained a base model on sunny scenes first, and then employed it as a teacher to train another network for weather scenarios with a specifically designed loss function. SSD[21] may be the most similar work to ours because it also takes diffusion to solve RMDE tasks. However, it follows the distillation paradigm of MD4all and introduces an enhanced dataset generated by diffusion, even altering the scene itself when rendering weather or night enhancements.

Contrastive-based methods typically improve the model's robustness by ensuring consistency between the depth predictions of a clear image and its augmented versions. Robust-Depth[26] introduces the semi-augmented warp and bi-directional contrast, which ensures the consistency assumption and improve the accuracy. WeatherDepth considers the domain gap between clear scenes and complex scenes and solves this problem by applying a gradual adaptation scheme based on curriculum contrastive learning. Additionally, EC-Depth[30] combines contrast and distillation with double-stage training. The first stage uses contrast, creating a KL divergence-like contrast to align the depth of harsh scenes, and the second stage uses distillation, introducing EMA for teacher-student joint learning.

## 3 METHOD

In this section, we will introduce the foundational knowledge of the self-supervised MDE and diffusion-based MDE (Sec. 3.1), the improvements to enhance the stability of the baseline (Sec. 3.2), the 'trinity' contrast paradigm (Sec. 3.3), the multi-level contrasting scheme (Sec. 3.4), and the two-stage training strategy (Sec. 3.5). An overview of the whole framework is shown in Fig. 2.

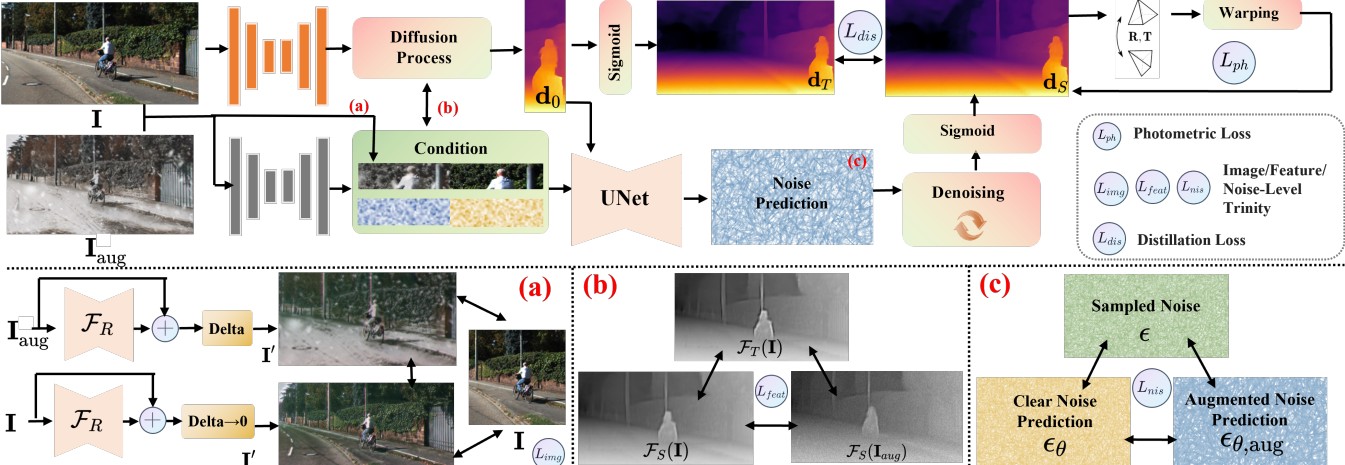

**Figure 2: The training framework of D4RD is depicted at the upper part of the image. Below that, multi-level trinity learning (*i.e.* images, depth features, and noise prediction) are presented through (a), (b), and (c), respectively.**

## 3.1 Preliminaries

**Self-Supervised MDE** uses an auxiliary image $I_a$ from the neighboring frame to constrains the output depth with a view-synthesis training pattern. Denoting the MDE model as $\mathcal{F} : I \rightarrow d \in \mathbb{R}^{W \times H}$, we can reach the warped target frame $I_{a \rightarrow t}$ with:

$$I_{a \rightarrow t} = I_a \langle \text{proj}(d, \mathcal{T}_{t \rightarrow a}, K) \rangle, \qquad (1)$$

where $\mathcal{T}_{t \rightarrow a}$ denotes the relative camera poses obtained from the pose network, and K denotes the camera intrinsics. Afterwards, we can compute the photometric reconstruction loss between $I$ and $I_{a \rightarrow t}$ to constrain the depth:

$$L_{ph} = \omega_1 \frac{1 - \text{SSIM}(I, I_{a \rightarrow t})}{2} + \omega_2 \|I - I_{a \rightarrow t}\|. \qquad (2)$$

In our implementation, DR4D takes the semi-augmented synthesis from Robust-Depth[26] to replace Eq. 1 with:

$$I'_{a \rightarrow t} = I_a \langle \text{proj}(d_{aug}, T_{t \rightarrow a}, K) \rangle, \qquad (3)$$

where $d_{aug}$ is the depth map inferred form the augmented images and $I'_{a \rightarrow t}$ will replace $I_{a \rightarrow t}$ in Eq. 2.

**Diffusion-based MDE** redefines the depth estimation task as a conditional denoising process. Given the image $I \in \mathbb{R}^{W \times H \times 3}$ and the corresponding depth distribution $D$, we can add the Gaussian noise $\epsilon$ to get the noisy sample $d_\tau$:

$$d_\tau = \sqrt{\bar{\alpha}_\tau} D + \sqrt{1 - \bar{\alpha}_\tau} \epsilon, \quad \epsilon \sim \mathcal{N}(0, I), \qquad (4)$$

and iteratively reverse it by removing the predict-added noise $\epsilon_\theta$:

$$\epsilon_\theta(d_{\tau-1} | d_\tau, c) := \mathcal{N}\left(\mu_\theta(d_\tau, \tau, c), \sigma_\tau^2 n\right), c = \mathcal{F}_f(I). \qquad (5)$$

Here $\tau$ consists of $\mathbb{T}$ steps, $\bar{\alpha}_\tau := \prod_{s=1}^{\tau} 1 - \beta_s$, and $\{\beta_1, \ldots, \beta_{\mathbb{T}}\}$ is the variance noise schedule for $\mathbb{T}$ steps [11]. The $\mathcal{N}(\cdot; \cdot)$ stands for gaussian sampling, $\mathcal{F}_f(\cdot)$ is a feature extractor, and $\sigma_\tau^2 n$ is typically set to $\mathbf{0}$ to make the estimation process deterministic.

Obviously, if $\epsilon_\theta$ is completely consistent with the $\epsilon$, we can reverse the depth result precisely, so we have the classical training loss:

$$L_{ddim} = \mathbb{E}_{d_0, \epsilon \sim \mathcal{N}(0,1), \tau \sim \mathcal{U}(T)} \|\epsilon - \epsilon_\theta(d_{\tau-1} | d_\tau, c)\|_2^2, \qquad (6)$$

where $\mathcal{U}(\cdot)$ is the uniformly sampling. For self-supervised learning, D4RD adopts the pseudo-GT diffusion as used in MonoDiffusion [28], which introduces a pre-trained model $\mathcal{F}_T$ to infer the pseudo-depth $d_T$ and it will replace $D$ in Eq. 4.

## 3.2 Stable Depth with Diffusion Models

**Pseudo-depth knowledge distillation.** To improve the performance, MonoDiffusion[28] adopted a weighted $L_1$ loss as the distillation constraint on the predicted depth. The weights are calculated based on the photometric difference using a fixed threshold. However, such a fixed threshold may stick the network to a performance bottleneck. To this end, we replace the fixed filter with an adaptive one:

$$M = \left[\min_a ph\left(I, I_{a \rightarrow t}^T\right) < \frac{\lambda}{\text{epoch}}\right], \qquad (7)$$

where $\lambda$ is a constant set to 1.5, and $I_{a \rightarrow t}^T$ is the warped target image using $d_T$. This dynamic weight initially enables D4RD to converge with the entire depth map and subsequently filters out the less accurate regions to reduce the negative effects brought by inaccurate pseudo-depth labels. Moreover, we find that BerHu loss[17] yields a lower error than the $L_1$ loss. It imposes a greater penalty on pixels with higher errors using $L_2$ loss, while maintaining the advantage of L1 loss at small errors. In short, we conclude with a dynamically weighted BerHu loss as our constraint for knowledge distillation:

$$L_{dis} = M \odot L_{berhu}(d_{S,aug}, d_{T,clr}), \qquad (8)$$

where $d_{T,clr}$ and $d_{S,aug}$ denote the predicted result of the clear or augmented images from $\mathcal{F}_T$ and $\mathcal{F}_S$, respectively.

**Outlier depth removal.** As mentioned before, since the iterative denoising process may produce unnatural depths (*e.g.*, negative values), previous methods[14] intend to implement diffusion in the

latent space:

$$\mathbf{d}_\tau = \sqrt{\overline{\alpha}_\tau}\mathcal{V}(\mathbf{d}) + \sqrt{1 - \overline{\alpha}_\tau}\epsilon, \mathbf{d} = \mathcal{V}^{-1}(\mathbf{d}_0), \quad (9)$$

where $\mathcal{V}(\cdot)$ is a pre-trained variational-auto-encoder (VAE) and $\mathcal{V}^{-1}(\cdot)$ is the reverse decoder. This manner undoubtedly introduces extra calculations. In this paper, we simplify the above solution through a non-linear activation function (and its inverse):

$$\mathbf{d}_\tau = \sqrt{\overline{\alpha}_\tau}\mathbf{sigmoid}^{-1}(\mathbf{d}_T) + \sqrt{1 - \overline{\alpha}_\tau}\epsilon, \mathbf{d} = \mathbf{sigmoid}(\mathbf{d}_0), \quad (10)$$

where $\mathbf{sigmoid}(\cdot)$ will transform the outputs into a limited range $[0, 1]$ and our experiments show that this simple manner can effectively remove the depth outliers. Eventually, D4RD adopts Eq. 10 in the denoising manner.

**Feature-image joint condition.** Our experiments have revealed that using the concatenation of the input image and its depth feature as the diffusion condition can bring better performance. It can be elaborated as:

$$\mathbf{c} = \mathbf{I} \oplus \mathcal{F}_f(\mathbf{I}), \quad \mathcal{F}_f(\mathbf{I}) \in \mathbb{R}^{1 \times H \times W}, \quad (11)$$

where $\oplus$ represents the channel-dim concatenation. We visualize the feature map in Fig. 4 to analyze the reasons and reveal that $\mathcal{F}(\mathbf{I})$ is a coarse 'depth map' with some additional incorrect edges. In other words, $\mathcal{F}_f$ focuses solely on depth-relevant information, and the combination in Eq. 11 provides higher-level contextual and semantic information, which helps the diffusion architecture in understanding the scenes, predicting noises, and ultimately generating better estimations.

### 3.3 Trinity Contrastive learning

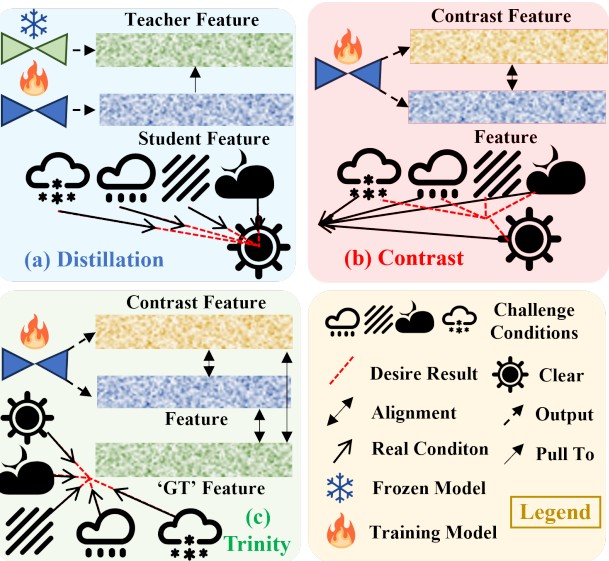

**Figure 3: Visual comparisons among three types of robust learning methods. Our trinity contrast method is more close to our expected effect and achieve better actual performance..**

The proposed trinity contrastive scheme integrates both knowledge distillation and contrastive learning. For the first one, the

general pattern is:

$$L_{dis} = |\mathcal{F}_S(\mathbf{I}_{aug}) - \underline{\mathcal{F}_T(\mathbf{I})}|, \quad (12)$$

where underline denotes the predicted from a frozen model. However, Fig. 3(a) has shown that this strategy struggles to teach $\mathcal{F}_S$ when the augmented image $\mathbf{I}_{aug}$ has a significant domain difference. Moreover, there is an evident performance upper bound for $\mathcal{F}_S$. For the latter, the usual form is:

$$L_{cst} = |\mathcal{F}(\mathbf{I}) - \mathcal{F}(\mathbf{I}_{aug})|. \quad (13)$$

As shown in Fig. 3 (b), without further guidance, simply gathering the depth map can easily lead to network collapse [4]. The self-supervised methods [26, 35] solve this by providing additional photometric guidance, but this guidance is not accurate enough and requires twice the training time (Eq.2 is calculated twice for $\mathbf{I}_{a \to t}$ and $\mathbf{I}'_{a \to t}$ separately).

In diffusion models, we discovered that in Eq. 4, the added noise $\epsilon$, which is sampled from $\mathcal{N}(0, I)$, can serve as a perfect guiding anchor for the estimated noise $\epsilon_\theta$ in Eq. 6 and do not require any extra annotation. Let $\epsilon_\theta$ and $\epsilon_{\theta,aug}$ represent the noise prediction for the clear image $\mathbf{I}$ and augment image $\mathbf{I}_{aug}$. Once these noises are identical and accurate, the final denoising depth will demonstrate consistency and robustness. Hence, as shown in Fig. 2(c), we reformulate a novel noise-level trinity contrast scheme as:

$$L_{nis} = \eta_1 \|\epsilon_{\theta,aug} - \epsilon_\theta\|_2^2 + \eta_2(\|\epsilon_{\theta,aug} - \epsilon\|_2^2 + \|\epsilon_\theta - \epsilon\|_2^2), \quad (14)$$

where $\eta_1$ and $\eta_2$ are set to 0.5 and 1, respectively . With the natural and perfect sampled noise label, the proposed trinity can stably gather the pair of estimated noises toward an optimal anchor position. We also experimentally prove its superiority to the other two traditional strategies in Section 4.4.

### 3.4 Multi-level Contrast

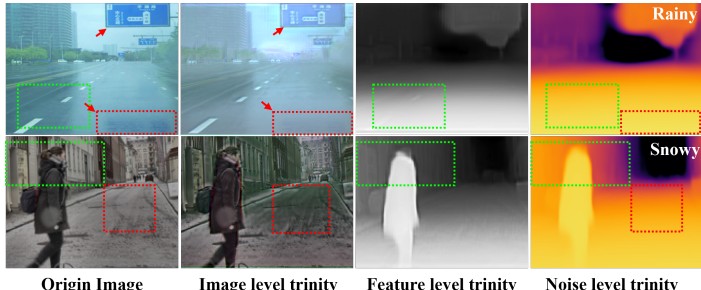

Origin Image     Image level trinity     Feature level trinity     Noise level trinity

**Figure 4: Visual results of each level trinity. Compared to the origin image, as shown in the highlight regions in the red dashed rectangle, the image level trinity can assist in handling water surface artifacts and ground snow. The feature level trinity builds a coarse depth map with some wrong edges but the noise trinity fixes them all. Better viewed when zooming in.**

Prior studies[13] have revealed that the burden of handling complex conditions mainly concentrates on a specific part of the network. Such an unbalanced burden may hinder the potential of

**Table 1: Quantitative results on WeatherKITTI and KITTI dataset. For the error-based metrics , the lower value is better; and for the accuracy-based metrics , the higher value is better. The best and second best results are marked in bold and underline.**

| Method | WeatherKITTI | | | | | | | KITTI | | | | | | |
|---|---|---|---|---|---|---|---|---|---|---|---|---|---|---|
| | AbsRel | SqRel | RMSE | RMSElog | $a_1$ | $a_2$ | $a_3$ | AbsRel | SqRel | RMSE | RMSElog | $a_1$ | $a_2$ | $a_3$ |
| MonoDiffusion[28] | 0.133 | 0.960 | 5.212 | 0.213 | 0.828 | 0.945 | 0.977 | 0.103 | 0.718 | 4.411 | 0.178 | 0.894 | 0.965 | 0.984 |
| MonoViT[36] | 0.120 | 0.899 | 5.111 | 0.200 | 0.857 | 0.953 | 0.980 | 0.099 | 0.708 | 4.372 | 0.175 | **0.900** | 0.967 | 0.984 |
| Robust-Depth[26] | 0.107 | 0.791 | 4.604 | 0.183 | 0.883 | 0.963 | 0.983 | 0.100 | 0.747 | 4.455 | 0.177 | 0.895 | 0.966 | 0.984 |
| EC-Depth[30] | 0.110 | 0.790 | 4.744 | 0.185 | 0.875 | 0.960 | 0.983 | 0.100 | 0.689 | 4.315 | 0.173 | 0.896 | 0.967 | **0.985** |
| Robust-Depth*[26] | 0.103 | 0.806 | 4.517 | 0.180 | 0.894 | 0.964 | 0.983 | 0.100 | 0.776 | 4.440 | 0.177 | 0.900 | 0.965 | 0.983 |
| WeatherDepth[31] | 0.103 | 0.738 | 4.414 | 0.178 | 0.892 | 0.965 | **0.984** | 0.099 | 0.698 | 4.330 | 0.174 | 0.897 | 0.967 | 0.984 |
| EC-Depth*[30] | 0.102 | 0.762 | 4.400 | 0.177 | 0.895 | **0.967** | 0.984 | 0.098 | 0.732 | 4.326 | 0.174 | **0.902** | **0.968** | 0.984 |
| D4RD | **0.099** | **0.688** | **4.377** | **0.174** | **0.897** | 0.966 | 0.984 | **0.097** | **0.665** | **4.312** | **0.171** | 0.900 | 0.967 | **0.985** |

robust algorithms. To this end, as shown in Fig. 2, we further extend the noise-level 'trinity' approach to more generic targets (*e.g.*, feature and image levels), through which we distribute the responsibility of robust perception across the whole network.

**Feature level.** We first extend it to the depth feature level, which exploits the robust ability of feature network $\mathcal{F}$. However, at this time, we lack the guidance as perfect as the added noise. Instead, as depicted in Fig. 2 (b), we adopt a teacher model $\mathcal{F}_T$ to get the suboptimal guiding label and build the feature-level 'trinity' loss as:

$$L_{feat} = \omega_1 \|\mathcal{F}_S(\mathbf{I}) - \mathcal{F}_S(\mathbf{I}_{aug})\|_1 + \\ \omega_2(\|\mathcal{F}_T(\mathbf{I}) - \mathcal{F}_S(\mathbf{I})\|_1 + \|\mathcal{F}_T(\mathbf{I}) - \mathcal{F}_S(\mathbf{I}_{aug})\|_1), \quad (15)$$

where $\omega_1$ and $\omega_2$ are empirically set to 1 and 0.5, respectively.

**Image level:** In addition to noise and feature, we also implement an image-level trinity contrast by setting the guiding label as the clear image $\mathbf{I}$. As depicted in Fig. 2(a), we design a simple CNN network that performs down-up sampling (denoted as $\mathcal{F}_R$) to get the enhanced image via:

$$\begin{cases} \mathbf{I}'_{aug} = \mathcal{F}_R(\mathbf{I}_{aug}) + \mathbf{I}_{aug} \\ \mathbf{I}' = \mathcal{F}_R(\mathbf{I}) + \mathbf{I} \end{cases} . \quad (16)$$

Then the image-level trinity is formulated as:

$$L_{img} = \beta_1(\|\mathbf{I}'_{aug} - \mathbf{I}\|_1 + \|\mathbf{I}' - \mathbf{I}\|_1) + \beta_2\|\mathbf{I}'_{aug} - \mathbf{I}'\|_1, \quad (17)$$

where the values of $\beta_1$ and $\beta_2$ are same as those of $\omega_1$ and $\omega_2$.

As shown in Fig. 4, through three trinity contrasts from different levels, the stresses of robust perception are evenly distributed among three parts of the network. They encourage the model to recognize and resist various types of scene degeneration at multiple stages, thus boosting the overall prediction reliability.

## 3.5 Two stage training

**Motivations.** We implement a two-step training strategy for two reasons. First, the proposed multi-level trinity contrastive paradigm requires the condition labels from the teacher model, but it does not exist initially. On the other hand, the teacher model $\mathcal{F}_T$ from the second stage can integrate the historical knowledge of stage one, yielding more accuracy and reliable pseudo-depth.

**The first training phase.** As shown in Fig. 2, We feed the clear image $\mathbf{I}_{cst}$ and its augmentation $\mathbf{I}$ as a mini-batch to the model and obtain their depth feature maps. Then we concat them

with their corresponding images with Eq. 11 to get the $\mathbf{c}_{cst}$ and $\mathbf{c}$. Next, both of them will be delivered to diffusion UNet to predict $\epsilon_\theta/\epsilon_{\theta,\text{aug}}$ and calculate the noise-level trinity loss $L_{nis}$. Note only $\epsilon_{\theta,\text{aug}}$ will be used for the denoising process and $L_{dis}$, $L_{ph}$ will be imposed on $\mathbf{d}$, which implies the proposed trinity will not double the computational burden as the traditional contrastive learning. With an extra edge-aware constraint $L_e$ from [10], we finally conclude the total loss at the first stage:

$$L_{stage1} = L_{nis} + L_{dis} + L_{ph} + \rho L_e, \quad (18)$$

where $\rho$ is empirically set to 1e-3.

**The second training phase.** Building upon the optimization objective of stage one, we add the $L_{feat}$ and $L_{img}$ to build the multi-level contrasts as mentioned in Section 3.4. The total loss of the second stage can be written as:

$$L_{stage2} = L_{nis} + L_{dis} + L_{ph} + \rho L_e + \theta(L_{feat} + L_{img}), \quad (19)$$

where $\theta$ is empirically set to 0.5.

## 4 EXPERIMENT

### 4.1 Dataset

**WeatherKITTI**[31] is built on the KITTI dataset and includes 6 kinds of weather augmentation. We train D4RD on this dataset because, compared with other synthetic datasets[21, 26], it is more realistic in simulating real-world complex scenarios. WeatherKITTI shares the same train/validate/test splits as KITTI. Therefore, for training, we follow Zhou's split[38] (containing 19,905 training images and 2,212 validation images), and for testing, we adopt Eigen's spilt[7] with 697 test images.

**KITTI-C**[16] is a synthetic dataset building for the RoboDepth Competition. In total, the benchmark contains 18 types of perturbations with 5 levels of corruption magnitudes, most of which are stronger than WeatherKITTI. We use it to fairly compare D4RD with their competition champions[30] to further demonstrate our robust performance.

**DrivingStereo**[33] is a large-scale real-world dataset. It contains four subsets of images under four weather conditions (*i.e.*, fog, cloudy, rainy, and sunny), with each subset containing 500 images.

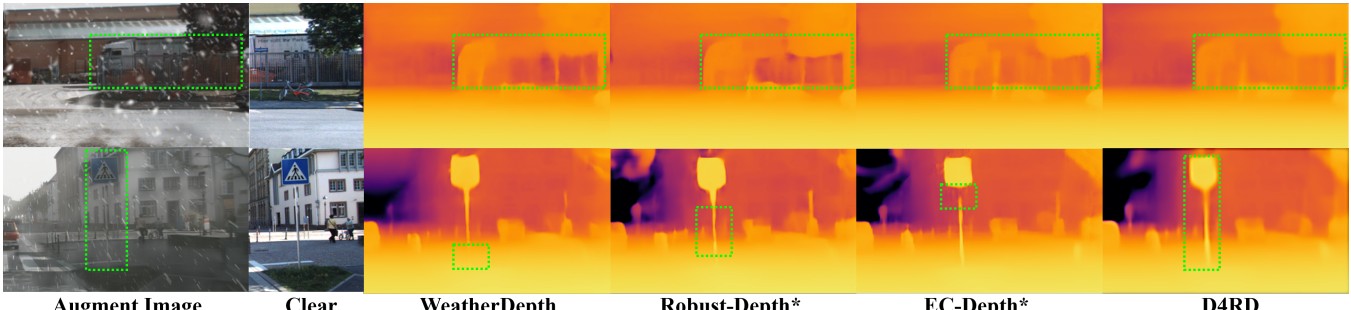

|Augment Image | Clear | WeatherDepth | Robust-Depth* | EC-Depth* | D4RD|

**Figure 5: Qualitative results for WeatherKITTI dataset. We compare D4RD with the current SoTA RMDE methods in the adverse rain and snow subsets. The part marked with 'Clear' is the correspondence sunny image (processed for more clarity). Regions with prominent differences are highlighted using dashed boxes.**

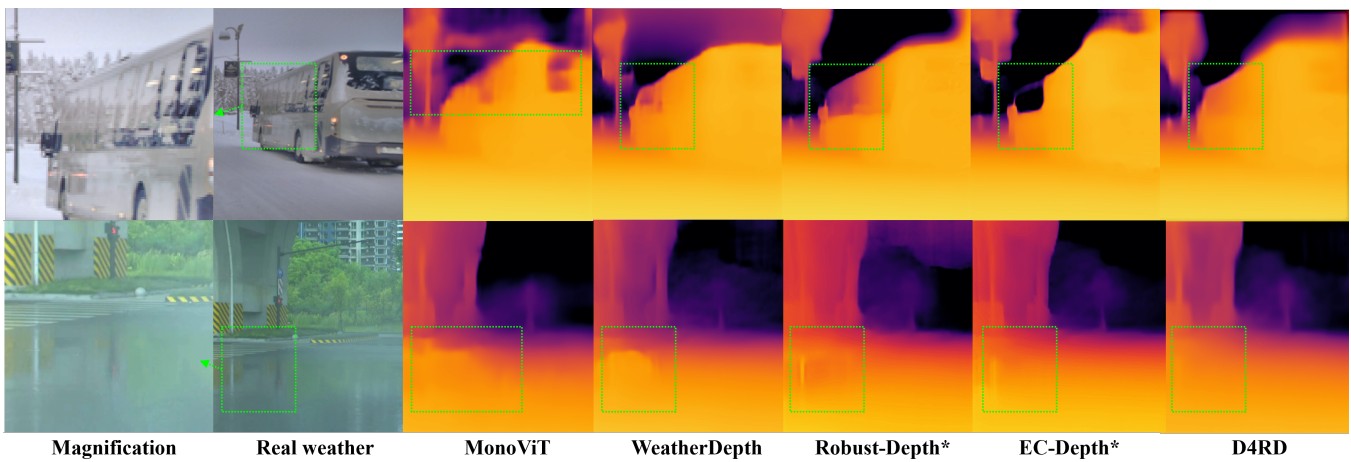

|Magnification | Real weather | MonoViT | WeatherDepth | Robust-Depth* | EC-Depth* | D4RD|

**Figure 6: Qualitative results for real weather dataset. Besides the RMDE methods, we also compare D4RD with the standard method MonoViT[36] in snowy and rainy environments to show our method's practicality. The part marked with Magnification is processed for more clarity.**

We test RMDE models on this dataset to characterize their ability to handle complex real-world conditions.

**Dense**[1] is one of the few real snowy datasets and it has approximately 1,500 snowy images in total. We introduce this benchmark to test the snowy scenes. To ensure consistency with the above dataset, we followed [31] and sampled 1 image every 3 to obtain 500 images as test data. To generate accurate depth GT in challenging snowy conditions, we use the DROR[2] algorithm to remove inaccurate depth readings from LiDARs affected by snowflakes. In addition, a small portion of distorted areas has been trimmed, resulting in a final resolution of $1880 \times 924$. We will explain the reasons for not using the snowy part of CADC dataset in the supplementary materials.

## 4.2 Implementation Details

In this paper, D4RD is based on MonoDiffusion[28], which is currently the only diffusion-based self-supervised MDE pipeline that combines the diffusion architecture with Lite-Mono[34]. To maintain consistency with previous RMDE researches[26, 30, 31], we replace Lite-Mono with MonoViT[36] in our D4RD.

D4RD is implemented in PyTorch and trained on a single NVIDIA RTX 3090. We use the AdamW optimizer[20] and other similar hyperparameters as MonoViT. The batch size is set to 6, and the training process runs 30 epochs in each stage. Considering the resolution difference across the datasets, we resize them to $640 \times 192$ for both training and testing. For the diffusion process, we follow MonoDiffusion[28] and set it with 1000 training steps and 20 inference steps.

For fair comparisons, we retrained Robost-Depth[26] and EC-Depth[30] on the WeatherKITTI dataset. Furthermore, to demonstrate D4RD's ability to handle robust scenes, we also retrain our model with the same training data used by [30], named D4RD$^{\dagger}$, to compare the performance on the KITTI-C dataset.

## 4.3 Quantitative and Qualitative Comparison

Here, we adopt the same evaluation metrics and errors up to 80m for all comparisons as [10]. Please refer to the supplementary for more details. The comparative methods include the baseline MonoDiffusion, traditional RMDE models' baseline MonoViT[36], previous SoTA RMDE models WeatherDepth[26], EC-Depth[30], Robust-Depth[26], and their retained models. Besides, [31] and [31] apply their approaches to various baselines, we want to clarify that we choose the one that uses MonoViT[36]. In [30], they take a two-stage training pipeline and we select the second stage model for the comparison.

**WeatherKITTI result.** We show detailed comparative experiments on the KITTI and WeatherKITTI datasets in Table 1, where our method outperforms other SoTA methods in all metrics by a large margin. Specifically, D4RD demonstrates a 25.6% and 17.5% decrease in AbsRel errors compared to the baselines(MonoDiffusion and MonoViT) under challenging conditions (WeatherKITTI). Furthermore, its performance either exceeds or closely matches that of these baselines in standard conditions (KITTI). The qualitative results are shown in Fig. 5. From the regions highlighted in green boxes, the compared schemes are affected by weather particle occlusion and blurs. In contrast, D4RD can robustly mine the consistent depth plane of the truck body and accurately estimate the correct contour of the road sign.

**Real Weather Scenes Results.** To validate the practicality of D4RD, we conduct zero-shot evaluations on real-world challenging datasets. As shown in Table 2 (a), D4RD still exhibits the best performance on the out-of-distribution clear dataset, demonstrating that our model has a stronger zero-shot ability than previous RMDE methods. Subsequently, we test the robustness of our model using the most common real-world changing conditions, including cloudy, rainy, foggy, and snowy scenes. As Table 2 shows, D4RD consistently demonstrates the SoTA performance in every scenario (b-e), and (f) further reports that our average performance has improved by a level compared to other methods in AbsRel( 1.41 to around 1.45), which strongly proves the superiority and practicality of our method. Fig. 6 displays the qualitative results of our method on real rainy and snowy environments. It can be observed that our method distinguishes the snow scene from the white bus (its windows even reflect some snow scenes) and better identifies artifacts caused by water surface reflections.

**KITT-C Result.** In addition to dealing with weather conditions, another role of RMDE is to resist image corruption and perturbations. The first track of RoboDepth competitions[16] focused on solving this problem. The solution from EC-Depth[30] won 1st place among dozens of participants and Table 3 shows its giant improvement from MonoViT. However, D4RD$^{\dagger}$ further defeats the EC-Depth solution by simply replacing the training dataset with EC-Depth's dataset, which powerfully proves that D4RD is currently the best RMDE framework.

## 4.4 Ablation Study

In this section, we conduct detailed ablation studies on WeatherKITTI and real-world benchmarks to demonstrate the effectiveness of the proposed components.

**Table 2: Zero-shot evaluation on the Dense and DrivingStereo dataset.**

| Method | AbsRel | SqRel | RMSE | RMSElog | $a_1$ | $a_2$ | $a_3$ |
|---|---|---|---|---|---|---|---|
| (a)DrivingStereo Sunny | | | | | | | |
| MonoDiffusion[28] | 0.162 | 1.738 | 7.535 | 0.220 | 0.799 | 0.936 | 0.975 |
| MonoViT[36] | 0.150 | 1.615 | 7.657 | 0.211 | **0.815** | 0.943 | 0.979 |
| Robust-Depth[26] | 0.152 | 1.574 | 7.293 | 0.210 | 0.812 | 0.944 | 0.979 |
| EC-Depth[30] | 0.151 | **1.436** | 7.213 | 0.209 | 0.808 | 0.944 | 0.979 |
| Robust-Depth*[26] | 0.152 | 1.651 | 7.369 | 0.212 | 0.810 | 0.944 | 0.978 |
| WeatherDepth[31] | 0.155 | 1.562 | 7.356 | 0.213 | 0.803 | 0.941 | 0.978 |
| EC-Depth*[30] | 0.155 | 1.538 | 7.301 | 0.211 | 0.804 | 0.941 | **0.980** |
| D4RD | **0.149** | 1.437 | 7.121 | **0.207** | **0.815** | 0.946 | 0.980 |
| (b)DrivingStereo Cloudy | | | | | | | |
| MonoDiffusion[28] | 0.155 | 1.853 | 7.822 | 0.214 | 0.807 | 0.939 | 0.978 |
| MonoViT[36] | **0.141** | 1.626 | 7.550 | 0.201 | **0.831** | 0.948 | 0.981 |
| Robust-Depth[26] | 0.148 | 1.781 | 7.472 | 0.204 | 0.825 | 0.947 | 0.981 |
| EC-Depth[30] | 0.147 | 1.561 | 7.301 | 0.201 | 0.825 | 0.947 | **0.983** |
| Robust-Depth*[26] | 0.147 | 1.749 | 7.486 | 0.205 | 0.823 | 0.946 | 0.982 |
| WeatherDepth[31] | 0.144 | **1.549** | 7.349 | 0.201 | 0.822 | **0.949** | 0.983 |
| EC-Depth*[30] | 0.150 | 1.655 | 7.517 | 0.204 | 0.819 | 0.945 | 0.982 |
| D4RD | **0.141** | 1.560 | 7.271 | **0.198** | 0.830 | 0.948 | **0.983** |
| (c)DrivingStereo Rainy | | | | | | | |
| MonoDiffusion[28] | 0.196 | 2.629 | 10.546 | 0.254 | 0.691 | 0.905 | 0.973 |
| MonoViT[36] | 0.175 | 2.136 | 9.618 | 0.232 | 0.730 | 0.931 | 0.979 |
| Robust-Depth[26] | 0.167 | 2.019 | 9.157 | 0.221 | 0.755 | 0.938 | 0.982 |
| EC-Depth[30] | 0.162 | 1.746 | **8.538** | 0.212 | 0.755 | **0.947** | **0.986** |
| Robust-Depth*[26] | 0.173 | 2.154 | 9.452 | 0.226 | 0.733 | 0.934 | 0.982 |
| WeatherDepth[31] | **0.158** | 1.833 | 8.837 | **0.211** | 0.764 | 0.945 | 0.985 |
| EC-Depth*[30] | 0.162 | 1.810 | 8.792 | 0.215 | 0.744 | 0.943 | 0.985 |
| D4RD | **0.158** | **1.722** | 8.584 | **0.208** | **0.773** | 0.946 | 0.985 |
| (d)DrivingStereo Foggy | | | | | | | |
| MonoDiffusion[28] | 0.128 | 1.540 | 8.687 | 0.191 | 0.831 | 0.955 | 0.986 |
| MonoViT[36] | 0.109 | 1.204 | 7.760 | 0.167 | 0.870 | 0.967 | 0.990 |
| Robust-Depth[26] | 0.105 | 1.132 | 7.273 | 0.158 | 0.882 | 0.974 | 0.992 |
| EC-Depth[30] | **0.105** | 1.061 | 7.121 | 0.155 | 0.880 | 0.974 | **0.994** |
| Robust-Depth*[26] | 0.111 | 1.240 | 7.536 | 0.163 | 0.873 | 0.971 | 0.992 |
| WeatherDepth[31] | 0.110 | 1.195 | 7.323 | 0.160 | 0.878 | 0.973 | 0.992 |
| EC-Depth* [30] | 0.111 | 1.177 | 7.315 | 0.160 | 0.870 | 0.974 | 0.993 |
| D4RD | **0.105** | **1.061** | 7.102 | **0.154** | **0.883** | **0.975** | **0.994** |
| (e)Dense Snowy | | | | | | | |
| MonoDiffusion[28] | 0.173 | 2.169 | 10.029 | 0.273 | 0.744 | 0.898 | 0.955 |
| MonoViT[36] | 0.162 | 2.063 | 9.787 | 0.262 | 0.762 | 0.904 | 0.960 |
| Robust-Depth[26] | 0.157 | 1.992 | 8.945 | 0.240 | 0.786 | 0.923 | 0.971 |
| EC-Depth[30] | 0.155 | **1.866** | 8.828 | 0.237 | 0.780 | 0.922 | **0.972** |
| Robust-Depth*[26] | 0.157 | 2.050 | 8.951 | 0.243 | 0.785 | 0.921 | 0.969 |
| WeatherDepth[31] | 0.157 | 2.000 | 9.021 | 0.243 | 0.781 | 0.919 | 0.968 |
| EC-Depth*[30] | **0.154** | 1.984 | 8.806 | **0.235** | 0.782 | 0.923 | **0.972** |
| D4RD | **0.154** | 1.868 | 8.780 | 0.236 | **0.789** | **0.925** | **0.972** |
| (f)Average | | | | | | | |
| MonoDiffusion[28] | 0.163 | 1.986 | 8.924 | 0.230 | 0.774 | 0.927 | 0.973 |
| MonoViT[36] | 0.147 | 1.729 | 8.474 | 0.215 | 0.802 | 0.939 | 0.978 |
| Robust-Depth[26] | 0.146 | 1.700 | 8.028 | 0.207 | 0.812 | 0.945 | 0.981 |
| EC-Depth[30] | 0.144 | 1.534 | 7.800 | 0.203 | 0.810 | 0.947 | **0.983** |
| Robust-Depth*[26] | 0.148 | 1.769 | 8.159 | 0.210 | 0.805 | 0.943 | 0.981 |
| WeatherDepth[31] | 0.145 | 1.628 | 7.977 | 0.206 | 0.810 | 0.945 | 0.981 |
| EC-Depth*[30] | 0.146 | 1.633 | 7.946 | 0.205 | 0.804 | 0.945 | **0.983** |
| D4RD | **0.141** | **1.530** | 7.772 | **0.201** | **0.818** | **0.948** | **0.983** |

**Table 3: Evaluation on the KITTI-C dataset.**

| Method | AbsRel | SqRel | RMSE | RMSElog | $a_1$ | $a_2$ | $a_3$ |
|---|---|---|---|---|---|---|---|
| MonoViT[36] | 0.161 | 1.292 | 6.029 | 0.247 | 0.768 | 0.915 | 0.964 |
| Robust-Depth[26] | 0.123 | 0.957 | 5.093 | 0.202 | 0.851 | 0.951 | 0.979 |
| EC-Depth[30] | 0.111 | 0.807 | **4.561** | 0.185 | 0.874 | 0.960 | 0.983 |
| D4RD$^{\dagger}$ | **0.108** | **0.778** | 4.652 | **0.183** | **0.880** | **0.961** | **0.983** |

**Table 4: Ablation study on stable depth with diffusion. All experiments are implemented in stage one and the meaning of the abbreviation is as follows. PDE: Pseudo-depth distillation enhancement; ODR: Outlier depth removing; FIC: Feature-image joint condition.**

| Benchmark | ID | PDE | ODR | FIC | Trinity | AbsRel | SqRel | RMSE | RMSElog | $a_1$ | $a_2$ | $a_3$ |
|-----------|----|----|----|----|---------|--------|-------|------|---------|-------|-------|-------|
| WeatherKITTI | 1 | | | | | 0.106 | 0.780 | 4.548 | 0.181 | 0.888 | 0.964 | 0.983 |
| | 2 | ✓ | | | | 0.104 | 0.775 | 4.502 | 0.179 | 0.892 | 0.965 | 0.983 |
| | 3 | ✓ | ✓ | | | 0.103 | 0.708 | 4.433 | 0.178 | 0.891 | 0.965 | **0.984** |
| | 4 | ✓ | ✓ | ✓ | | 0.101 | 0.712 | 4.423 | 0.177 | 0.895 | 0.965 | **0.984** |
| | 5 | ✓ | ✓ | ✓ | ✓ | **0.100** | **0.696** | **4.375** | **0.175** | **0.896** | **0.966** | **0.984** |

**Table 5: Ablation study on contrast manner and level. All experiments are implemented in stage second and the meaning of the abbreviation is as follows. NIS: the noise level contrast; DF: the depth feature level contrast; IMG: the image level contrast. Real Weather shows the average test result across all kinds of real weather conditions in Tab.2**

| Benchmark | ID | NIS | DF | IMG | Method | AbsRel | SqRel | RMSE | RMSElog | $a_1$ | $a_2$ | $a_3$ |
|-----------|----|----|----|----|--------|--------|-------|------|---------|-------|-------|-------|
| WeatherKITTI | 1 | | | | | 0.100 | 0.696 | 4.375 | 0.175 | 0.896 | 0.966 | 0.984 |
| | 2 | ✓ | | | Trinity | 0.100 | 0.700 | 4.367 | 0.175 | 0.897 | 0.966 | 0.984 |
| | 3 | ✓ | ✓ | | Trinity | **0.099** | 0.698 | **4.364** | 0.175 | **0.898** | 0.966 | 0.984 |
| | 4 | ✓ | ✓ | ✓ | Distill | 0.100 | 0.704 | 4.384 | 0.175 | 0.896 | 0.966 | 0.984 |
| | 5 | | ✓ | ✓ | Contrast | 0.100 | 0.712 | 4.425 | 0.176 | 0.895 | 0.965 | 0.984 |
| | 6 | ✓ | ✓ | ✓ | Trinity | **0.099** | **0.688** | 4.377 | **0.174** | 0.897 | **0.966** | **0.984** |
| Real Weather | 7 | | | | | 0.143 | 1.566 | 7.853 | 0.202 | 0.815 | 0.947 | 0.982 |
| | 8 | ✓ | | | Trinity | **0.141** | 1.540 | 7.851 | **0.201** | 0.817 | **0.948** | **0.983** |
| | 9 | ✓ | ✓ | | Trinity | **0.141** | 1.535 | 7.850 | 0.202 | 0.817 | 0.947 | 0.982 |
| | 10 | ✓ | ✓ | ✓ | Distill | 0.145 | 1.588 | 7.894 | 0.204 | 0.812 | 0.946 | 0.981 |
| | 11 | | ✓ | ✓ | Contrast | 0.142 | 1.538 | 7.835 | 0.202 | 0.815 | **0.948** | 0.982 |
| | 12 | ✓ | ✓ | ✓ | Trinity | **0.141** | **1.530** | **7.772** | **0.201** | **0.818** | 0.948 | **0.983** |

**Effectivness of Diffusion Enhancement.** Table 4 shows the effectiveness of each enhancement based on the first stage of training. We take the MonoDiffusion (replaced with MonoViT) as the baseline (ID 1). Firstly, the distillation enhancement (ID 2) is added and produces a significant improvement. This gain is reasonable because, compared with MonoDiffusion which only considers clear scenes, the depth estimation in complex scenes typically requires more powerful distillation. Next, we add outlier depth removal (ID 3), which brings a notable improvement in RMSE and SqRel metrics because the unnatural depth values have more significant impact on the second-order indicators. Finally, we improved the condition (ID 4). With the introduction of image context information, the diffusion-based model further shows an improvement in the $a_1$ metric representing estimation stability.

**Effectiveness of Trinity Contrast.** As reported in Table 4 (ID 5), the incorporation of trinity in the first stage model produces overall performance gains in all aspects. Meanwhile, in Table 5, we can compare the performance of three paradigms using IDs 4-6 and IDs 10-12. The results show that direct distillation did not enhance the performance and even led to a decrease in stability (RMSE, $a_1$). For the contrastive method, although we applied the constraint of Eq. 2 to the contrastive objects, directly contrasting the predicted noises would lead to network crashes (*i.e.*, in which all predicted depth values are zeros, which are not included in Table 5). Therefore, we still maintain the noise constraint from Eq. 6, only taking the feature and image levels comparison, but there is only a minor improvement. The proposed trinity contrast

is the only approach that significantly improves convergence on WeatherKITTI and demonstrates increased robustness and accuracy on real datasets.

**Effectiveness of Multi-Level Contrasts** The comparison of ablation with multi-level contrasts is presented in Table 5 (IDs 2,3,6 and IDs 8,9,12). While the performance of different models is close in the WeatherKITTI dataset, in real-world weather situations, we notice that the results become more reliable as we gradually include the feature and image level trinity. The enhancements in these metrics, coupled with Fig. 4, apparently prove the effectiveness of the multi-level contrast.

## 5 CONCLUSION

In this paper, we enhance the stability and convergence of diffusion-based MDE and further introduce a novel robust depth estimation framework named D4RD, which integrates a customized contrastive learning scheme. This method combines the strength of distillation with contrastive learning and repurposes the sampled noise in the diffusion forward process, creating a "trinity" contrastive mode, thereby improving the accuracy and robustness of noise prediction. Furthermore, we extend the noise level's trinity to the more general feature and image levels, constructing a multi-level trinity scheme to balance the robust perception pressure across different model components. Through extensive quantitative and qualitative experiments, we demonstrate the effectiveness of D4RD against various architectures and its superior performance over existing SoTA solutions.

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
