# OpenReview forum: "Digging into contrastive learning for robust depth estimation with diffusion models"
_acmmm.org/ACMMM/2024/Conference — MM2024 Poster_

### Official Review · Reviewer_rLfb · 2024-05-18

**Rating:** 4
**Confidence:** 3

**Summary:**

This work presents an approach called Diffusion for Robust Depth (D4RD) to enhance monocular depth estimation (MDE). D4RD introduces a diffusion-based MDE framework with a multi-level (image, feature, noise) trinity contrastive scheme designed to enhance robustness. The framework utilizes sampled noise from the forward diffusion process as a label to guide contrastive learning and knowledge distillation.

Experiment demonstrates state-of-the-art performance on synthetic corruption datasets and real-world weather conditions.

**Strengths:**

1. The technical description of the methods is thorough and clear. Implementation details are well described.

2. The motivation to utilize diffusion model on the trinity paradigm is clear. The method aims to overcome the limitation of knowledge distillation methods which is to overcome the upper bound of the student model's performance.

3. The trinity contrastive learning scheme is somewhat novel compared to other existing contrastive learning strategies.

4. Extensive experiments show the effectiveness of the method.

**Limitations:**

1. The memory cost during and training time with respect to other methods are not stated. The two-stage strategy can potentially introduce higher computational resource to implementation.

2. The noise level contrast seems to be one of the contribution. However from the empirical results of the ablation studies, the performance gain could possibly be a subset of the performance gain from depth level contrast and image level contrast.

3. Implementation details states that D4RD is based on MonoDiffusion with MonoViT replacing Lite-Mono. I think the most fair and straightforward way to demonstrate effectiveness of the framework is to train the original MonoDiffusion with and without the proposed method and compare the results.

**Suitability:**

2

---

### Official Review · Reviewer_n89p · 2024-05-24

**Rating:** 4
**Confidence:** 3

**Summary:**

The paper introduces D4RD, a framework for robust depth estimation that integrates a custom contrastive learning scheme with diffusion models. It enhances the stability and convergence of diffusion-based monocular depth estimation (MDE) by utilizing sampled noise as a natural reference for guiding noise prediction. D4RD also employs a multi-level 'trinity' contrastive scheme to distribute the robust perception burden across the network, demonstrating superior performance over existing solutions in challenging conditions.

**Strengths:**

1. Extensive experiments and comparisons with state-of-the-art methods demonstrate D4RD's effectiveness, providing confidence in its real-world applicability.
2. The use of sampled noise as a reference for contrastive learning is innovative and provides a natural and unsupervised way to guide the learning process.
3. The multi-level trinity scheme smartly distributes the robust perception responsibility, potentially enhancing the model's ability to handle complex scenes.

**Limitations:**

1. The method's reliance on diffusion models might introduce additional complexity and computational cost compared to traditional MDE approaches.
2. The effectiveness of the trinity contrastive scheme may be highly dependent on the quality of the teacher model, which could be a limitation if the teacher model is not accurate.
3. Authors doe not provide any analysis of the model's parameter count and computational effort, including inference speed, which must be considered in real-life applications.
4. The proposed method seems to be somewhat close to the suboptimal method, can the authors further analyze why?
5. The starting point of this paper is depth estimation in bad weather, however, the related works section does not explore any adverse weather restoration methods.

**Suitability:**

2

---

### Official Review · Reviewer_5Ts7 · 2024-05-26

**Rating:** 4
**Confidence:** 1

**Summary:**

The paper introduces an approach to enhance the robustness of depth estimation by incorporating multi-level contrastive learning into the MotionDiffusion model. This involves contrasting images and their augmented versions, noise estimations for both the image and its augmented counterpart within the diffusion model, and features of the image and augmented image from both the student model and between the teacher and student models. The authors also mention additional changes such as a revised loss function in distillation, outlier removal in latent space, and feature-image joint optimization. The effectiveness of the proposed network is validated through experiments conducted on four datasets and comparisons with several baselines.

**Strengths:**

The idea of imposing a contrastive constraint within the diffusion model using noise as a supervisory signal is intriguing and potentially impactful.

The authors conducted thorough experiments using four datasets and compared the proposed model with several baseline models, providing solid and persuasive results.

**Limitations:**

1: Network Structure Illustration:

The overall structure of the network is not well-illustrated in Figure 2, and the corresponding descriptions in the paper are confusing and poorly correlated. It is unclear which components correspond to the student or teacher network.
There is ambiguity regarding the "diffusion process". If it refers to the inverse generation stage of the diffusion model, the input should be some initial noise or intermediate denoised state in the latent space. The role of the encoder-decoder for outlier removal is also unclear in the figure.
The term "d_0" appears directly after the diffusion process and is then fed to the UNet for noise prediction, but it is unclear whether this represents the predicted image. The term "Delta" in Figure (a) is not explained, and there is no clear connection between Figures (b) and (c) and the specific components in the overall network structure.

2: Complexity and Clarity:

The network's complexity, combining numerous components and loss terms, makes it challenging to follow. The loss terms for the diffusion model are not clearly identified.

Two-stage training is used. Which components parts are trained in the first stage? After first-stage training, will the components weight fixed or trained again? Whether the weights are trained from start or from the one from first stage? When should the first stages be stopped?

The noise supervision part of the contrastive components in the diffusion model raises concerns. Specifically, it is unclear whether the same noise epsilon is injected into both the clear and augmented images, and whether it is correct to assert that epsilon_\theta should equal epsilon_{\theta, aug}. This part requires more detailed explanation.

The metrics measuring the performances of algorithms in the experiments are not explained.

3: Inconsistent Notations:

The notations throughout the paper are inconsistent and confusing. For instance, \theta in equations (6) and (19) is inconsistently used, and epsilon sometimes appears with \theta and sometimes without it. The clear image is sometimes denoted as I and other times as I_{cst}, while its augmentation is inconsistently referred to as I or I_{aug}. Additionally, the notation for \tau and T in diffusion is confusing when compared to their use in denoting Target (T) and Teacher model (T).

4: Lack of Comparative Results:

Regarding the three additional improvements (revised loss in distillation, outlier removal, and feature-image joint optimization), there are no comparative results provided to show the effectiveness of these changes. It would be helpful to see results before and after these improvements to understand their impact better.

**Suitability:**

3

---

### Meta-Review · Area_Chair_S98k · 2024-06-24

**Recommendation:** Accept (Poster)
**Confidence:** 5

**Metareview:**

This submission received three borderline accept ratings, with all three reviewers consistently leaning towards acceptance. As no final rating was provided, the Area Chair reviewed the paper further and decided to accept it. For the camera-ready version, the authors are encouraged to provide a complexity analysis of the proposed solution.